# Characterizing the Physicochemical Properties of Two Weakly Basic Drugs and the Precipitates Obtained from Biorelevant Media

**DOI:** 10.3390/pharmaceutics14020330

**Published:** 2022-01-29

**Authors:** Miao Zhang, Bin Wu, Shudong Zhang, Lin Wang, Qin Hu, Dongyang Liu, Xijing Chen

**Affiliations:** 1School of Basic Medicine and Clinical Pharmacy, China Pharmaceutical University, Nanjing 211198, China; 3119090253@stu.cpu.edu.cn; 2Drug Clinical Trial Center, Peking University Third Hospital, Beijing 100191, China; 3Institute of Medical Innovation, Peking University Third Hospital, Beijing 100191, China; 4NMPA Key Laboratory for Research and Evaluation of Generic Drugs, Beijing Institute for Drug Control, Beijing 102206, China; redwizard2008@gmail.com (B.W.); hx@bidc.org.cn (S.Z.); wanglin@bidc.org.cn (L.W.); huqin@bidc.org.cn (Q.H.)

**Keywords:** precipitate, physicochemical properties, biorelevant media, permeability, weakly basic insoluble drugs

## Abstract

Generally, some weakly basic insoluble drugs will undergo precipitate and redissolution after emptying from the stomach to the small intestinal, resulting in the limited ability to predict the absorption characteristics of compounds in advance. Absorption is determined by the solubility and permeability of compounds, which are related to physicochemical properties, while knowledge about the absorption of redissolved precipitate is poorly documented. Considering that biorelevant media have been widely used to simulate gastrointestinal fluids, sufficient precipitates can be obtained in biorelevant media in vitro. Herein, the purpose of this manuscript is to evaluate the physicochemical properties of precipitates obtained from biorelevant media and active pharmaceutical ingredients (API), and then to explore the potential absorption difference between API and precipitates. Precipitates can be formed by the interaction between compounds and intestinal fluid contents, leading to changes in the crystal structure, melting point, and melting process. However, the newly formed crystals have some advantageous properties compared with the API, such as the improved dissolved rate and the increased intrinsic dissolution rate. Additionally, the permeability of some precipitates obtained from biorelevant media was different from API. Meanwhile, the permeability of rivaroxaban and Drug-A was decreased by 1.92-fold and 3.53-fold, respectively, when the experiments were performed in a biorelevant medium instead of a traditional medium. Therefore, the absorption of precipitate may differ from that of API, and the permeability assay in traditional medium may be overestimated. Based on the research results, it is crucial to understand the physicochemical properties of precipitates and API, which can be used as the departure point to improve the prediction performance of absorption.

## 1. Introduction

Physiologically based pharmacokinetic (PBPK) models have been widely used to predict the pharmacokinetic characteristics of compounds, but they cannot always capture the absorption process, leading to low confidence (greater than 2-fold) in the prediction performance [1,2]. In fact, the oral absorption of some insoluble drugs is susceptible to gastrointestinal (GI) conditions, e.g., the pH in the GI tract, the bile micelle concentration under both fasted and fed states, and so on, which can impact the solubility and permeability of drugs, increasing the difficulty of accurately estimating the absorption characteristic of compounds [3,4,5,6].

Additionally, oral absorption is a complex process in the GI tract, especially concerning salts and weakly basic insoluble drugs. Generally, the weakly basic insoluble compounds show good solubility in the acidic gastric environment through ionization, but the advantage dissipates when the less soluble unionized compounds are formed in the near-neutral intestinal microenvironment. Nucleation will occur when a large number of the unionized compounds are continuously concentrated to form a supersaturation solution, followed by crystallization and precipitate [7,8]. Moreover, the precipitate can be redissolved again and cross the intestinal barrier after the full absorption of the dissolved unionized compound [9]. Therefore, one compound can undergo dissolution, supersaturation, precipitation, and redissolution before transporting across the barrier tissue into the systemic circulation in theory [10]. Many studies have been carried out on the supersaturation and precipitate of the weakly basic insoluble drugs in vivo, and the phenomenon of precipitate does exist in vivo [11,12,13].

To improve the prediction accuracy of absorption for insoluble compounds, the PBPK absorption model focuses on the dynamic process of the compounds in the GI tract. Although many factors, such as supersaturation, precipitation time, and dissolution profiles [14,15], have been considered in the development of the PBPK absorption model for poorly soluble drug candidates, it is unclear whether the absorption process of the new crystals is the same as that of the API in the intestinal microenvironment. Nucleation and crystallization occur in the intestinal environment, which may be affected by the chemical bond or the properties of the intraluminal components in intestinal fluid [16,17]. Some studies have reported that the crystallization structure is different from the API [7,11]. Therefore, it is crucial to characterize the physicochemical properties of crystals to better understand the absorption differences between precipitates and API.

Considering that biorelevant media, simulating physicochemical parameters and components of intestinal fluid, is commonly used in dissolution tests to evaluate the potential dissolution processes of formulations in vivo in advance [15], it is an alternative method for gathering sufficient crystals in biorelevant media in vitro. Therefore, evaluation of physicochemical properties in vitro is a workable plan. Herein, the crystals of model drugs were collected in biorelevant media and their physicochemical properties were evaluated systematically by X-ray diffraction, scanning electron microscopy, Fourier transform infrared spectroscopy, differential scanning colorimetry, permeability assay, and so on. Through these explorations, we hope to shed light on the potential absorption difference between API and the redissolved precipitates and provide some points for reasonably improving the prediction accuracy of absorption.

## 2. Materials and Methods

### 2.1. Materials

Rivaroxaban (pKa 13.6, log *p* 1.90, purity ≥ 98%) and Drug-A (pKa 10.41, log *p* 2.102, purity ≥ 98%) were selected as weakly basic model drugs in this manuscript. Both of them were obtained as gifts from sponsors. Biorelevant powder, a commercial product of premixed bile salts and phospholipids, was purchased from Biorelevant.com Ltd. (Surrey, UK). Chemicals were purchased from Sinopharm Chemical Reagent Co. Ltd. (Shanghai, China), except for methanol, acetonitrile, and potassium bromide obtained from Merck (Merck KGaA, Darmstadt, Germany). All reagents were analytical grade and were used without further purification. Parallel artificial membrane permeability assay (PAMPA), acceptor sink buffer, and lipid solution were obtained from Pion Inc. (Billerica, MA, USA).

### 2.2. Equilibrium Solubility of API in Different Media

To select an appropriate method for the preparation of precipitates, the equilibrium solubility of model drugs was firstly investigated in various media, including pH 1.0 HCl solution, pH 5.0, pH 6.5, and pH 8.0 phosphate buffer solution (PBS), fasted state simulated intestinal fluid (FaSSIF, Fa), and fed-state simulated intestinal fluid (FeSSIF, Fe) [18]. The excess sample was added to the medium, and the test of equilibrium solubility was performed with the stirring rate of 200 revolutions per minute (rpm) under 37 °C for 24 h. Sample concentration was detected every minute by Pion’s in situ fiber-optic UV dissolution monitoring system (Pion Scientific Instruments, Billerica, MA, USA), a real-time online detector. All measurements were performed in triplicate.

### 2.3. Preparation of Precipitates

The biorelevant media, including FaSSIF and FeSSIF media, were prepared based on the instructions from biorelevant.com. The preparation method of double diluted (0.5 × FeSSIF, 0.5 Fe) and double concentrated (2 × FeSSIF, 2 Fe) FeSSIF media was similar to FeSSIF medium, but the amount of biorelevant powder was reduced and increased to twice that of FeSSIF medium, respectively. The composition of the media is summarized in Table 1 [19]. The precipitates of the compound were prepared by the solvent shift method [20]. Rivaroxaban and Drug-A were first dissolved in a minimal amount of glacial acetic acid and acetonitrile, respectively. Then, the concentrated compound solutions were separately added into the biorelevant media (FaSSIF, 0.5 × FeSSIF, FeSSIF, and 2 × FeSSIF) and aqueous buffer media (FaSSIF buffer (FaB) and FeSSIF buffer (FeB)). The precipitate in each medium was obtained according to the solubility difference of model drugs in organic solvent and the biorelevant media/aqueous buffers. The precipitates were collected by vacuum suction filtration and washed three times with a few milliliters of distilled water. The samples were dried in a vacuum drying oven at 40 °C for 24 h and stored with desiccators until further analysis [21].

### 2.4. Recovery of Precipitates

The content of rivaroxaban and Drug-A in the corresponding crystal was determined with a known concentration solution, which was prepared by dissolving a certain amount of precipitate in a volumetric flack [22]. Then the absolute content of corresponding crystals was determined by high-performance liquid chromatography [23] (HPLC, Agilent Technologies Santa Clara, CA, USA) and compared with that of plain API solutions. All these tests were performed three times.

### 2.5. X-ray Diffraction of Precipitates and API

The diffraction patterns of API and precipitates that were obtained from the organic solvent, FaSSIF buffer, FeSSIF buffer, FaSSIF, 0.5 × FeSSIF, FeSSIF, and 2 × FeSSIF media, respectively, were analyzed by X-ray diffraction (XRD) using Rigaku ultima IV with CuKα radiation (λ = 0.154184 nm) operating at 40 KV and 40 mA. The pattern was recorded for angles (2θ) in the range of 0–45 degrees at a scanning rate of 4 °C/min [24].

### 2.6. Crystal Morphology of Precipitates and API

The surface morphology of different crystal structures was characterized by scanning electron microscopy (SEM, Zeiss SUPRA 55 VP FE-SEM; Carl-Zeiss Jena; Jena, Germany). All samples were coated with a moderate gold-palladium layer and then observed and recorded at an acceleration voltage of 10.0 kV [25].

### 2.7. Fourier Transform Infrared Spectroscopy

A Fourier transform infrared spectroscope (FTIS, Thermo Fisher Scientific Inc., Waltham, MA, USA) was used to evaluate the transition of functional groups of precipitates after the interaction between the compound and molecules in solutions. Samples were ground with spectroscopy grade KBr at the weight ratio of 1:50 and then compressed into a semitransparent pellet under the pressure of 20 tons for 30 s. The spectra signal was collected 16 times in the range of 400–4000 cm^−1^ with a resolution of 4 cm^−1^ for each sample [26]. The OMNIC 9.3 was used for FTIR data analysis.

### 2.8. Differential Scanning Calorimetry

Thermal analyses of various samples were carried out by differential scanning calorimetry (DSC Q20, TA Instruments, New Castle, DE, USA). Indium was firstly used to calibrate the instrument. Then, the samples were weighted accurately and placed in aluminum pans. The temperature was programmed at a rate of 10 °C/min with a starting temperature of 100 ℃. Nitrogen at a flow rate of 40 mL/min was used as purge gas [26].

### 2.9. Equilibrium Solubility of Precipitates and API in pH 6.5 Aqueous Buffer

To estimate the solution properties of different crystal structures in near neutral or slightly acidic environments, the equilibrium solubility of various precipitates and API in the pH 6.5 FaSSIF buffer medium (FaB) was investigated. An excess sample was added to the medium, and the test of equilibrium solubility was performed with the stirring rate of 200 revolutions per minute (rpm) under 37 °C for 24 h. Sample concentration was detected every minute by Pion’s in situ fiber-optic UV dissolution monitoring system. All measurements were performed three times.

### 2.10. Intrinsic Dissolution Rate of Precipitates and API

The intrinsic dissolution rate (IDR) of compounds with different crystal structures was measured by the disc method based on the literature [27,28,29,30]. About 10 mg of powder was loaded into the Mini-IDR compression system (Pion Inc., Billerica, MA, USA) and compressed at 100 psi (rivaroxaban) and 200 psi (Drug-A) for 1 min to obtain the miniaturized discs with a surface area of 0.071 cm^2^. The discs were inserted into stirrer-die assembly and placed into flat-bottomed glass vials for dissolution analysis. At the start of the experiments, the magnetic stirrer was immediately switched on with the speed of 200 rpm, and then 10 mL of preheated (37 °C) dissolution media (FaSSIF buffer medium, FeSSIF buffer medium, FaSSIF medium and FeSSIF medium) were added into the vials at a constant temperature. The concentration was determined directly in the dissolution media every 30 s using an in situ UV probe for 2 h. All experiments were performed in triplicate, and the IDR was calculated with the following equation [30]:(1)$$IDR=V\times \frac{dc}{dt}\times \frac{1}{A}$$
where *V* is the volume of medium in the glass vial; *dc/dt* is the slope of the straight line from the dissolution profile; *A* is the surface area of the disc.

### 2.11. Permeability Assay of Precipitates and API in Different Media

In order to simulate the permeation of compounds in different crystal structures in the GI tract, the stirring PAMPA method (Pion Inc., Billerica, MA, USA) was used to assess the passive transcellular permeability. The donor and acceptor chambers representing the GI tract and systemic circulation were separated by the artificial membrane (PVDF, polyvinylidenfluoride, 0.45 μm, 0.78 cm^2^). The membrane surface was coated with 20 μL of lipid, which was used to simulate the microenvironment of lipids in vivo. Then, 16 mL PBS/FaSSIF/FeSSIF medium with pH 5.0 was added into the donor chamber, and 16 mL sink buffer was added into the acceptor chamber. In order to simulate the redissolution progress of precipitate in the GI tract, a slight excess of crystalline powder was added into the donor chamber. Both chambers were stirred at 200 rpm at 37 °C, and the concentration of each chamber was detected by immersed UV-probes at 320 nm and 340 nm for rivaroxaban and Drug-A, respectively. The flux was defined as the amount of compound passing through the unit area perpendicular to its flow direction in unit time and was calculated by the following equation [31]:(2)$$J\left(t\right)=\frac{C_{t_{2}}-C_{t_{1}}}{A\left(t_{2}-t_{1}\right)}\times V$$
where *J*(*t*) is the flux of a compound; *C_t_* is the concentration of compound at *t*; *V* is the solution volume of the donor chamber; *A* is the area of the artificial membrane.

The correlation between *J* and the effective permeability (*P*_e_) can be described as follows [32]:(3)$$J=P_{e}\times \left(C_{d}-C_{a}\right)$$
where *J* is the flux of the compound; *C**_d_* is the concentration of the compound in the donor chamber; *C**_a_* is the concentration of the compound in the accepter chamber; *P_e_* is the effective permeability of the compound. Since the concentration of the compound in the receptor is very low at the beginning of the experiment, assuming that the concentration of API in the receptor chamber is zero, the equation can be simplified as follows [32]:(4)$$J=P_{e}\times c_{d}$$

### 2.12. Statistical Analyses

The statistical analyses regarding significance levels and correlation analysis were assessed with the GraphPad Prism Version 8.0 (GraphPad Software, San Diego, CA, USA), and a *p*-value < 0.05 was used to estimate the significance.

## 3. Results

### 3.1. Solubility of the API in Different Media

The equilibrium solubility of rivaroxaban and Drug-A in various aqueous buffers and biorelevant media was firstly tested. The detailed results are shown in Table 2. Because the change of pH value of the medium had no significant effect on the solubility of model drugs, the solvent shift method was used to prepare precipitates [20].

### 3.2. Recovery of Precipitates

The content of rivaroxaban and Drug-A in precipitates were accurately determined with HLPC. The absolute quantitative method for rivaroxaban refers to the published literature [23], and Drug-A refers to the declared data of new drug applications (Unpublished). The recoveries of all the test samples are within the range of 95–101%, indicating that there is little loss during the recrystallization. The detailed results are shown in Table 3.

### 3.3. X-ray Diffraction Results of Precipitates and API

XRD analyses are performed to identify crystalline phases, and the detailed results are shown in Figure 1. There is little effect of organic solvent on the characteristic peak of rivaroxaban and Drug-A, respectively. However, the media have different influences on the crystal structure. Aqueous media mainly affect the peak strength rather than the peak position, and the presence of bile salts in the medium can change the crystalline natures, as the characteristic diffraction peaks appear at 2θ equal to 3 and 9 degree for rivaroxaban, and 6, 32, and 41 degree for Drug-A. Moreover, changes in bile salts concentration can impact the relative strength of peaks. Therefore, the crystal lattice structure and lattice energy of compounds have been changed during the formation of precipitates.

### 3.4. Scanning Electron Microscopy Analyses

To investigate the possible microscopic morphology of crystals in GI tract, API and precipitates obtained from FaSSIF and FeSSIF media were characterized by SEM. The different rulers were selected to capture the complete crystal morphology based on the size difference of crystals. Figure 2A show the microscopic morphology of rivaroxaban (API). Clearly, there are some nearly spherical particles with the size range of 1–10 μm, and some small particles stick together to form larger particle clusters. Figure 2B present the morphology of precipitate (rivaroxaban) obtained from the FaSSIF medium. Some lamellar blocks with different sizes were recorded, and the thick sheet block consisted of many thin sheets with a thickness of several nanometers. The microscopic morphology of precipitate (rivaroxaban) obtained from the FeSSIF medium is shown in Figure 2C, and there exist some loose blocks with many irregular small particles adhered to the surface. Figure 2D (API of Drug-A) display many small sugar block particles with a clear surface contour, and the particle size is less than 1 μm. The microscopic morphology of precipitate (Drug-A) obtained from FaSSIF medium (Figure 2E) presents a similar morphology to that of precipitate (rivaroxaban) obtained from the FeSSIF medium (Figure 2C). Namely, some large block samples have many small particles adhered to the surface. Figure 2F present many long rod-shaped blocks with different sizes, and the largest size is up to dozens of microns. Overall, the precipitates obtained from FaSSIF and FeSSIF media have an obvious cuboid structure with fine particles attached to the surface. Moreover, the particle size of precipitates is larger than that of API.

### 3.5. Fourier Transform Infrared Spectroscopy Analysis

Since the crystalline properties of precipitates were different from those of API, FTIR analysis was used to determine whether the functional groups of compounds had changed in the microenvironment. Figure 3 show the FTIR spectra of APIs and precipitates of rivaroxaban and Drug-A. The characteristic peaks of the rivaroxaban related samples appear at the same absorption position, indicating that precipitates and API are still the same compounds with different crystal structures (Figure 3A). However, Drug-A related precipitates obtained from the media containing acetic acids, such as FeSSIF buffer, 0.5 × FeSSIF, FeSSIF, and 2 × FeSSIF media, have two new absorption peaks at the position of 1750 cm^−1^ and 1725 cm^−1^, which may be caused by the stretching of C=O bonds of saturated fatty acid monomers (acetic acid). In order to confirm the type of carbon–oxygen double bond, API and these precipitates were dissolved in acetonitrile and analyzed by mass spectrometry. Both the parent ions and product ions between API and the precipitates obtained from medium containing acetic acid are the same (Unpublished). Combined with the structural formula of Drug-A, the basic drug may interact with the acetic acid and form carboxylate.

### 3.6. Differential Scanning Calorimetry Analysis

Under the same conditions, there is no significant difference in melting endotherm among the precipitates of rivaroxaban (Figure 4A). While for Drug-A, the melting points and melting process are changed dramatically with precipitates obtained from FeSSIF buffer, 0.5 × FeSSIF, FeSSIF, and 2 × FeSSIF media. As shown in Figure 4B, the endothermic peak of precipitates obtained from organic solvent, FaSSIF buffer, and FaSSIF media is very sharp but precipitates obtained from FeSSIF buffer, 0.5 × FeSSIF, FeSSIF, and 2 × FeSSIF media display a smooth and shallow peak during the endothermic process, respectively. Changes in thermostability may be related to the newly formed carbon–oxygen double bond after the interaction between the Drug-A and acetic acid in the buffer medium. Hence, the composition of supersaturated solutions can influence the crystal structure and then affect the thermostability of the compound.

### 3.7. Equilibrium Solubility of Different Precipitates and API in the Same Medium

Thermodynamic properties, e.g., melting point and fusion enthalpy of crystals, and the interaction between solutes and compounds can jointly influence the equilibrium solubility of a compound in a specific medium. Hence, the equilibrium solubility of these precipitates was measured in FaSSIF buffer medium at pH 6.5. As shown in Figure 5A (rivaroxaban related), the dissolved rates of all the precipitates are higher than that of API, and the extent (achieved in about 4 h and 24 h) is almost the same. Interestingly, the dissolved rates (Figure 5B) for Drug-A and its precipitates are different. Some precipitates (Fa and 0.5 Fe in Figure 5B) dissolved more slowly and had less solubility than that of API in the first 8 h. Precipitates obtained from FeSSIF and 2 × FeSSIF media have high solubility at the beginning of dissolution, but then the solubility decreases. This phenomenon may be caused by the formation of supersaturation solution after rapid dissolution in the medium and then trend to the equilibrium solubility. Furthermore, for rivaroxaban and Drug-A, there is no statistical difference in equilibrium solubility between API and precipitates obtained from biorelevant media (FaSSIF, 0.5 × FeSSIF, FeSSIF, and 2 × FeSSIF). Therefore, the biorelevant precipitates mainly have the characteristic of faster dissolution than API, which may be caused by the improvement of wettability.

### 3.8. Intrinsic Dissolution Rate

Since the dissolved rate of biorelevant precipitates is faster than API, the intrinsic dissolution rate of samples was estimated to explore the impact of crystal structure on dissolution rate [30]. To ensure the consistency of experimental conditions, IDR is governed by the surface area of the compound in contact with the dissolution medium [28]. IDR of different samples was calculated according to Equation (1). Figure 6 exhibit the correlation between concentration and time in the experiment of IDR. The final results are shown in Figure 7. The crystal structure of rivaroxaban has a significant influence on IDR (*p* < 0.05). However, except for IDR of 0.5 Fe in FaSSIF buffer, there is no statistical difference in the IDR of Drug-A with different crystal structures in the same dissolution medium. In addition, the pH value and composition of the dissolution medium have great influences on the IDR of the same sample (*p* < 0.05). Therefore, precipitates obtained from biorelevant media may have some improved properties of dissolution, which are the pivotal points affecting the absorption of compounds.

### 3.9. Permeability Assay

Considering that the crystal structure, dissolved rate, and IDR were different among precipitates, permeability tests were performed with the PAMPA. The permeability is calculated according to Equations (2)–(4), and the results are shown in Figure 8. In Figure 8A, the permeability of some rivaroxaban related precipitates was larger than that of API at pH 5.0 PBS (donor chamber) (*p* < 0.05). When the dissolution medium was FeSSIF in the donor chamber, only the permeability of API (rivaroxaban) was significantly different from that of precipitates obtained from 0.5 × FeSSIF and FeSSIF media (*p* < 0.05). However, the permeability of API was larger than that of precipitates obtained from biorelevant media. As shown in Figure 8B, only the precipitate obtained from 0.5 × FeSSIF medium had different permeability from other samples when the donor chamber contained pH 5.0 PBS (*p* < 0.05). Therefore, under the same conditions, the crystal structure has limited influence on permeability. Nevertheless, the dissolution medium in the donor chamber is one of the factors affecting permeability. The permeability of rivaroxaban and Drug-A in the donor chamber containing aqueous medium is about 1.92-fold and 3.53-fold higher than that of samples in the donor chamber containing biorelevant medium, respectively. Actually, solubility and permeability are related to the polarity and non-polarity of the compound, respectively [33]. Although biorelevant media have good solubilization ability, the amount of bile salts is limited in its capacity to dissolve all compounds in the donor chamber. Theoretically, the hydrophobic groups of compounds can be wrapped by bile salts, and the hydrophilic groups are exposed to the medium. After the binding between the compound and the bile salts, the increasing molecular size and the hydrated layer will prevent the compounds from passing through the lipophilic permeable membrane. In addition, a medium with a high bile salts concentration (such as FeSSIF medium) can increase the binding of insoluble compounds to bile salts in the donor chamber, thus further reducing the permeability of the compounds. Therefore, there is a negative correlation between permeability and solubility.

## 4. Discussion

There are several ways to form a supersaturated solution in vivo. The supersaturated solution of some insoluble weak basic compounds can be triggered by the pH gradient from the stomach to the small intestine, resulting in the generation of precipitate and reduction of absorbable compounds, which has attracted extensive attention [34]. A supersaturated solution is a thermodynamically unstable system with high chemical potential, which drives the formation of crystallization/precipitate into an energetically favorable crystal structure. Part of the precipitate is excreted from the body with the intestinal peristalsis. When the concentration of the compound in the GI tract falls below the equilibrium solubility, the remaining precipitates are redissolved. Therefore, the complex precipitate kinetics of compounds in vivo play a key role in governing oral absorption. However, the limiting step of precipitate absorption is unknown. Absorption is related to the solubility and permeability of the substrate [35] and is determined by several factors, including physicochemical, formulation-relevant, and physiological factors [36]. Chemical groups of compounds may interact with intestinal fluid components during crystallization, resulting in different physicochemical properties between the newly formed crystals and API. Systematic study of the physicochemical properties of precipitates obtained from biorelevant media in vitro is the prerequisite for understanding the absorption mechanism of redissolved precipitates.

Precipitates obtained from the biorelevant media have many different physicochemical properties from that of API, but precipitates cannot be regarded as new active substances unless they demonstrate different safety and efficacy [37]. Actually, precipitates have some improved properties compared with the API, such as the rapid dissolution and the increase in intrinsic dissolution rate. These favorable changes may be related to the interaction between biorelevant media and compounds in the crystallization process so as to improve the wettability of compounds. Therefore, the absorption of redissolved precipitate may differ from API. Since the absolute amount of unabsorbed precipitate is unknown, this may be an important factor affecting the prediction performance of absorption. In addition, not all weakly basic compounds precipitate in the GI tract and not all precipitate events are equal, which depend on the solubility difference between the stomach and intestinal chambers, chemical groups, lattice energy, and so on. Hence, it is very important to determine whether precipitate can be formed in the GI tract through a biorelevant two-stage in vitro test [38] and then characterize the physicochemical properties of precipitates from biorelevant media.

Considering that changes in bile salts composition and content may lead to differences in crystallization trends of compounds [39], physicochemical properties of precipitates obtained from biorelevant media with different bile salts contents were investigated in this manuscript. The changes of dissolved rate, intrinsic dissolution rate, and melting point appear in different crystals of rivaroxaban and Drug-A, indicating that bile salt concentration will affect the absorption of compounds and may also be one of the factors leading to individual absorption differences. Bile secretion is related to food intake and fat content. Hence, the influence of fat content and the difference of absorption mechanism between precipitates and API should be considered when using the PBPK model to predict the absorption of compounds. In addition, bile salts such as endogenous surfactants play an important role in solubilizing and improving the rate of dissolution of insoluble compounds. With the rapid dissolution of compounds into the medium, supersaturated solutions exceeding the equilibrium solubility can be formed [40]. Meanwhile, bile salts can interact with drug molecules, modify the crystallization kinetics, prolong the duration of supersaturated behavior, and finally promote the absorption of compounds [41,42,43]. Therefore, bile salts have complex effects on solubility and permeability, and the permeability of compounds should be estimated in the biorelevant systems.

Many studies have focused on evaluating drug permeation under fed state conditions. However, exogenous compounds, such as intestinal fluid and bile salts under fed state, can induce detrimental changes to epithelial cell monolayer integrity [44], cytotoxic effects [45], cell viability [46], and so on. These disadvantageous changes make it more difficult to evaluate the permeability of compounds by cells experiments under fed states, and the accuracy of experimental results may be limited [47]. In addition, crystal structure and dissolution rate are the pivotal factors determining the solubility of compounds, and supersaturation solution has a high solute activity that can promote the penetration of the compound into the intestinal epithelial cells [10]. Therefore, adding the predissolved solution of compounds into the donor chamber may ignore the generation of supersaturation solution. Here, the crystal structures, bile salts, and other factors affecting drug dissolution are incorporated into the permeability evaluation of the PAMPA method by adding excess compounds into the donor chamber. Meanwhile, the experimental results are meaningful, as the permeability is negatively correlated with bile salt concentration. This phenomenon indicates that compounds may have different permeability under fasted and fed states in vivo, and traditional permeability evaluation under standard conditions cannot represent the permeability of compounds in vivo.

PBPK absorption models have been widely used to predict the effects of food on the exposure of compounds, with 50% of the observed food effects within the predefined boundary of 25% [1,2]. To accurately capture absorption characteristics, factors affecting the absorption of compounds in the model, such as dissolution rate and precipitate rate, are usually evaluated under the biorelevant media to simulate the physiological environment [48,49]. However, the absorption process of redissolved precipitates may differ from that of API due to differences in physicochemical properties, particularly permeability. Moreover, apparent permeability is a ubiquitous descriptor of drug absorption and a widely used input value of the PBPK model based on experimental results. Based on the research results in this manuscript, this seemingly simple parameter requires additional consideration in order to better understand the most appropriate value that should be used in the model. Although PAMPA can evaluate the effects of crystal structure and intestinal fluid on permeability, it is an artificial membrane, and the metabolism and transport of compounds in the GI tract are neglected. The permeability assay of rivaroxaban and Drug-A in this manuscript shows that the medium in the donor chamber is the pivotal factor affecting the apparent permeability. The correlation between apparent permeability in biorelevant media and the effective permeability of humans should be further studied. Hence, it is urgent to establish sophisticated experimental systems to evaluate the permeability of drugs with a high level of bio-relevance, which is also the focus of our next research.

## 5. Conclusions

Physicochemical properties are the key points affecting the absorption of compounds in vivo. The reabsorption of precipitates is an important part of the absorption of some insoluble, weakly basic oral drugs. Hence, evaluating the physicochemical properties of precipitates collected from biorelevant media is fundamental for a full understanding of compound absorption. The crystal structure, melting point, melting process, and fusion enthalpy are different between API and precipitates. The interactions between API and biorelevant media can affect the molecular structure of a certain compound, such as Drug-A. Nevertheless, precipitates have many physicochemical properties superior to API, including the rapid dissolution into the medium and the increasing intrinsic dissolution rate. In addition, the permeability of some precipitates is statistically different from that of API. Meanwhile, bile salts can reduce the permeability of rivaroxaban and Drug-A by 1.92-fold and 3.53-fold, respectively. Considering that the dissolved rate of the precipitates is improved, the permeability of rivaroxaban-related precipitates is smaller than that of API, and the permeability assessment in vitro overestimates that in vivo, more attention should pay for absorption of redissolved precipitates and reasonably evaluating the permeability of compounds with a high level of bio-relevance.

## Figures and Tables

**Figure 1 pharmaceutics-14-00330-f001:**
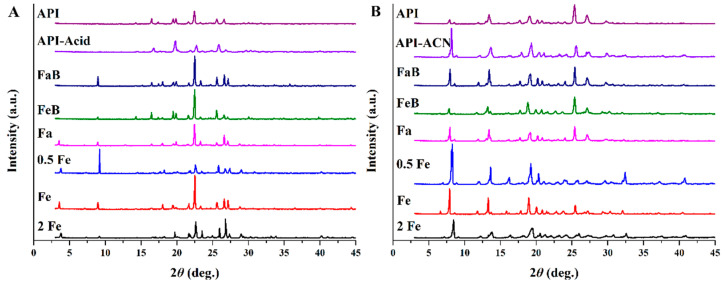
X-rays diffractogram of different samples. (**A**): Rivaroxaban; (**B**): Drug-A. (API-Acid/API-ACN, FaB, FeB, Fa, 0.5 Fe, Fe, and 2 Fe indicate that precipitate comes from the medium of organic solvent, FaSSIF buffer, FeSSIF buffer, FaSSIF, 0.5 × FeSSIF, FeSSIF, and 2 × FeSSIF, respectively).

**Figure 2 pharmaceutics-14-00330-f002:**
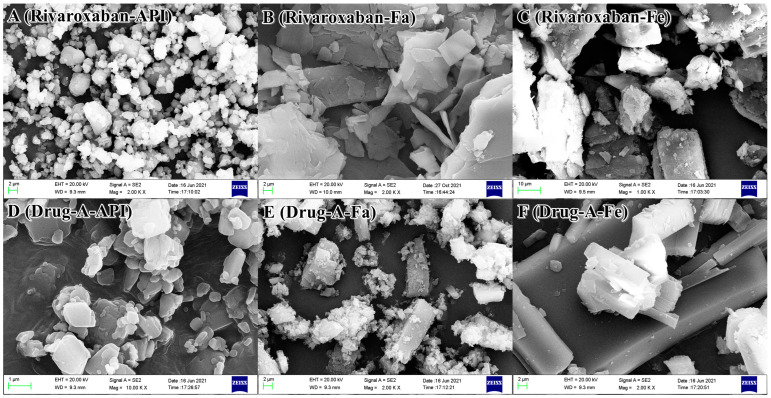
The microscopic morphology of API and precipitates. (**A**): API of rivaroxaban; (**B**) precipitate of rivaroxaban comes from FaSSIF medium; (**C**) precipitate of rivaroxaban comes from FeSSIF medium; (**D**): API of Drug-A; (**E**) precipitate of Drug-A comes from FaSSIF medium; (**F**) precipitate of Drug-A comes from FeSSIF medium).

**Figure 3 pharmaceutics-14-00330-f003:**
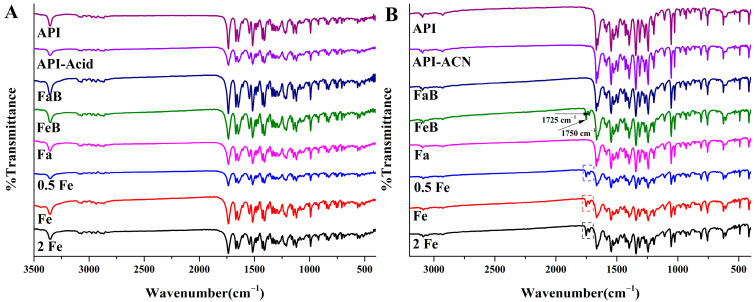
Fourier transform infrared spectrum of API and different precipitates. (**A**): Rivaroxaban; (**B**): Drug-A. (API-Acid/API-ACN, FaB, FeB, Fa, 0.5 Fe, Fe, and 2 Fe indicate that precipitate comes from the medium of organic solvent, FaSSIF buffer, FeSSIF buffer, FaSSIF, 0.5 × FeSSIF, FeSSIF, and 2 × FeSSIF, respectively).

**Figure 4 pharmaceutics-14-00330-f004:**
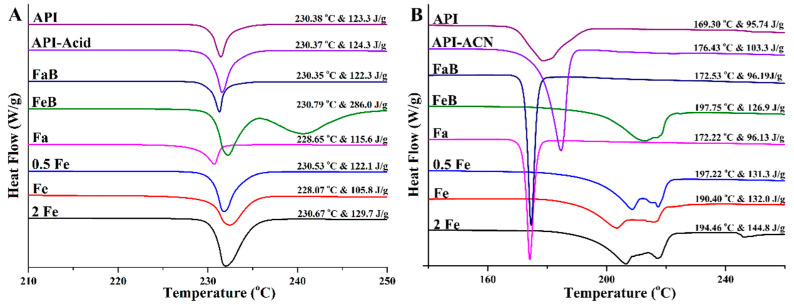
DSC thermogram of different precipitates. (**A**): Rivaroxaban; (**B**): Drug-A. (API-Acid/API-ACN, FaB, FeB, Fa, 0.5 Fe, Fe, and 2 Fe indicate that precipitate comes from the medium of organic solvent, FaSSIF buffer, FeSSIF buffer, FaSSIF, 0.5 × FeSSIF, FeSSIF, and 2 × FeSSIF, respectively).

**Figure 5 pharmaceutics-14-00330-f005:**
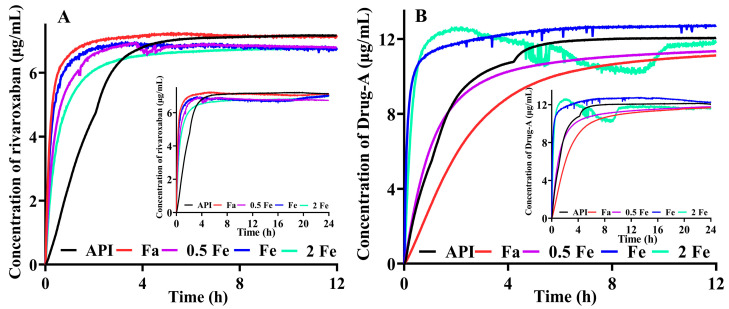
The dissolution profiles of different precipitates and API in the same dissolution medium. (**A**): Rivaroxaban; (**B**): Drug-A. (Fa, 0.5 Fe, Fe, and 2 Fe indicate that precipitate comes from the medium of FaSSIF, 0.5 × FeSSIF, FeSSIF, and 2 × FeSSIF, respectively).

**Figure 6 pharmaceutics-14-00330-f006:**
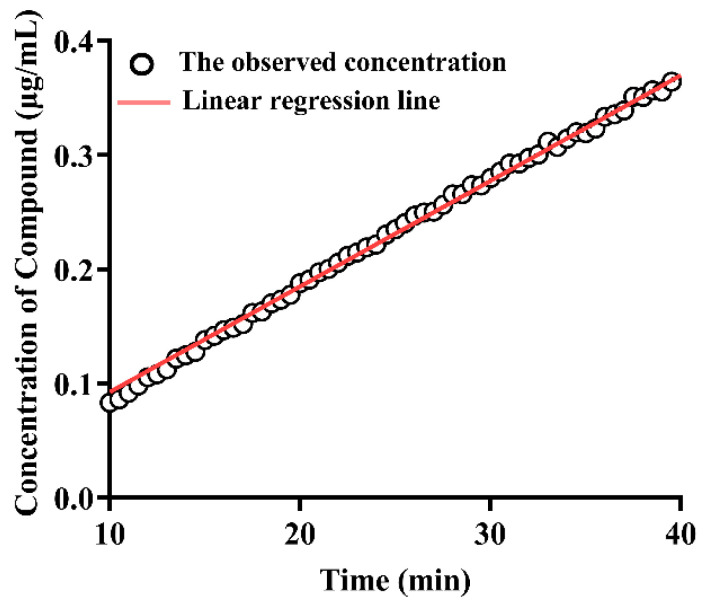
The correlation between concentration and time in the IDR experiments.

**Figure 7 pharmaceutics-14-00330-f007:**
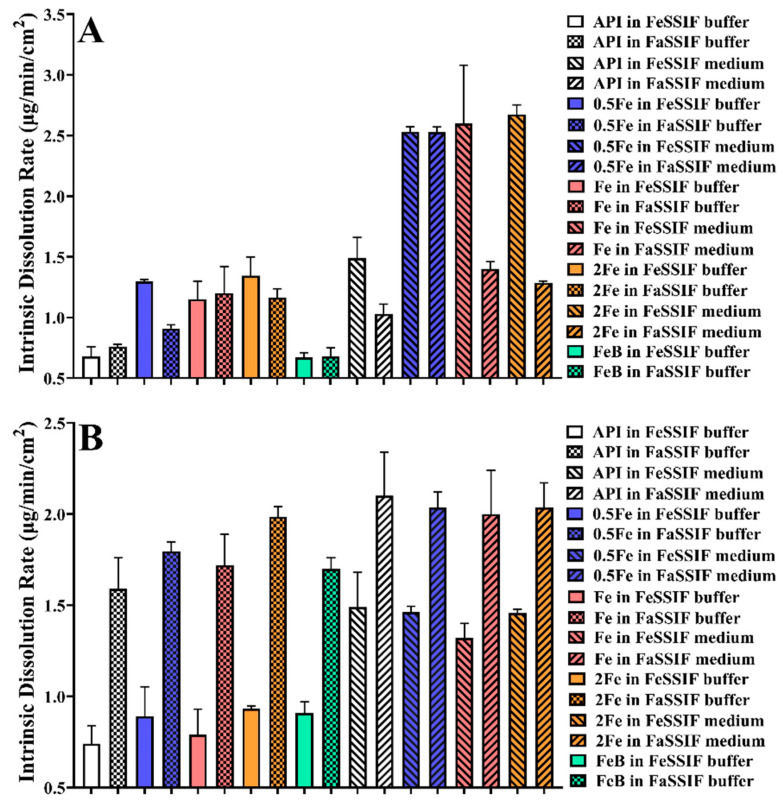
The intrinsic dissolution rate of different samples in various media. (**A**): Rivaroxaban; (**B**): Drug-A. (FeB, 0.5 Fe, Fe, and 2 Fe indicate that precipitate comes from the medium of FeSSIF buffer, 0.5 × FeSSIF, FeSSIF, and 2 × FeSSIF, respectively).

**Figure 8 pharmaceutics-14-00330-f008:**
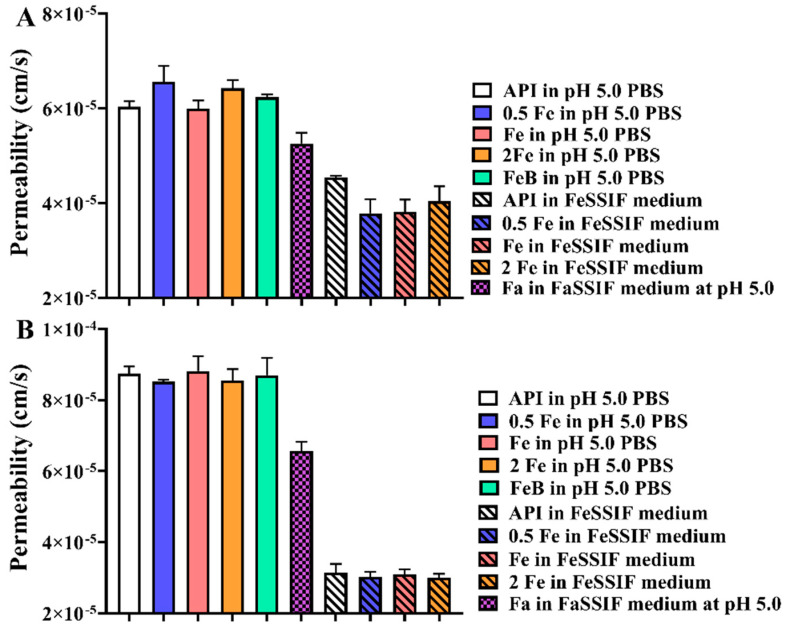
The permeability of different precipitates in different media. (**A**): Rivaroxaban; (**B**): Drug-A. (FeB, Fa, 0.5 Fe, Fe, and 2 Fe indicate that precipitate comes from the medium of FeSSIF buffer, FaSSIF, 0.5 × FeSSIF, FeSSIF, and 2 × FeSSIF, respectively).

**Table 1 pharmaceutics-14-00330-t001:** The composition of a liter of medium.

Media	NaOH	NaH_2_PO_4_	NaCl	CH_3_COOH	pH	Biorelevant Powder *
	(g)	(g)	(g)	(g)		(g)
FaSSIF buffer (FaB)	0.42	3.44	6.19	-	6.5	-
FeSSIF buffer (FeB)	4.04	-	11.87	8.65	5.0	-
FaSSIF (Fa)	0.42	3.44	6.19	-	6.5	2.24
FeSSIF (Fe)	4.04	-	11.87	8.65	5.0	11.2
0.5 × FeSSIF (0.5 Fe)	4.04	-	11.87	8.65	5.0	5.60
2 × FeSSIF (2 Fe)	4.04	-	11.87	8.65	5.0	22.4

* Using the same biorelevant powder to prepare FaSSIF, 0.5 × FeSSIF, FeSSIF and 2 × FeSSIF media.

**Table 2 pharmaceutics-14-00330-t002:** The solubility of API in different media. (*n* = 3).

Model Drugs	pH 1.0 HCl	pH 5.0 PBS	pH 6.5 PBS	pH 8.0 PBS	FaSSIF (μg/mL)	FeSSIF (μg/mL)
	(μg/mL)	(μg/mL)	(μg/mL)	(μg/mL)		
Rivaroxaban	6.84 (±0.20)	4.85 (±0.11)	5.85 (±0.02)	5.87 (±0.04)	6.17 (±0.13)	10.6 (±0.14)
Drug-A	42.9 (±0.47)	11.9 (±0.12)	11.6 (±0.17)	11.2 (±0.16)	12.3 (±0.30)	13.8 (±0.85)

**Table 3 pharmaceutics-14-00330-t003:** Recoveries of compounds from corresponding precipitates. (*n* = 3).

Model Drugs	Organic Solvent	FaSSIF Buffer	FeSSIF Buffer	FaSSIF	0.5 × FeSSIF	FeSSIF	2 × FeSSIF
Rivaroxaban	98.7% (±1.59%)	100% (±0.49%)	99.0% (±0.96%)	96.6% (±1.12%)	98.1% (±1.32%)	95.2% (±1.38%)	99.6% (±1.16%)
Drug-A	99.3% (±0.17%)	99.6% (±0.40%)	100% (±1.34%)	101% (±0.91%)	99. 8% (±1.01%)	97.3% (±0.99%)	98.2% (±0.10%)

## Data Availability

Raw and processed data are available from D.L. (liudongyang@bjmu.edu.cn) upon reasonable request.

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
