# Peer review of "Characterizing the Physicochemical Properties of Two Weakly Basic Drugs and the Precipitates Obtained from Biorelevant Media"

_pharmaceutics, 2022, doi:10.3390/pharmaceutics14020330_

Round 1

Reviewer 1 Report

The topic submitted is novel and adds significant research data to the existing field of research. The article is not very well articulated needs English language revisions and even formatting of the manuscript as per the MDPI guidelines. The manuscript needs to be checked for statistical significance especially fig 6 & 7. The introduction needs to be concise. Abstract and conclusion should include a sentence proposing the future direction of the present research. Also, the commercial aspects of how precipitations obtained from bio-relevant media industries can benefit.  Most of the methodology lacks citing the references from where the technique was adopted and used to conduct the study. The below are a few relevant article suggestions that authors are recommended to cite in the methodology, results /discussion section to justify and strengthen the research data obtained.  

International journal of pharmaceutics 511, no. 1 (2016): 680-687.

Gels 7, no. 3 (2021): 96

International journal of pharmaceutics 453, no. 1 (2013): 25-35.

International journal of nanomedicine 10 (2015): 321.

Journal of Pharmacy and Pharmacology 63, no. 3 (2011): 333-341.

Current drug delivery 13, no. 2 (2016): 211-220.

Author Response

Dear Reviewer,

Thank you for your letter and comments concerning our manuscript entitled “Characterizing the physicochemical properties of two weakly basic insoluble drugs and the precipitates obtained from biorelevant media” (Pharmaceutics-1520066).. Those comments are all valuable and very helpful for improving our paper. We have carefully studied the comments and have made appropriate corrections in line with the recommendations of reviewers. Changes are highlighted with yellow background in the revised version. The main corrections in the manuscript and the responses to the reviewers’ comments are as follows:

Reviewer 1

Comment 1: The topic submitted is novel and adds significant research data to the existing field of research. (1) The article is not very well articulated needs English language revisions and even formatting of the manuscript as per the MDPI guidelines. (2) The manuscript needs to be checked for statistical significance especially fig 6 & 7.

Reply: Thanks for your valuable comments.

(1) According to your suggestion, we have polished the language carefully and changes are highlighted with yellow background in the new manuscript.

(2) The statistical significance of some important data including Fig. 6 (Figure 7 in the new manuscript) and Fig. 7 (Figure 8 in the new manuscript) difference from was checked. We have revised it and the details as follows:

“The final results are shown in Figure 7. Crystal structure of rivaroxaban has the significant influence on IDR (P<0.05). However, except for IDR of 0.5 Fe in FaSSIF buffer, there is no statistic difference in IDR of Drug-A with different crystal structures in the same dissolution medium.”

“In Figure 8 A, the permeability of some rivaroxaban related precipitate was larger than that of API at pH 5.0 PBS (donor chamber) (P<0.05). When the dissolution medium was FeSSIF in the donor chamber, only the permeability of API (rivaroxaban) was significantly different from that of precipitates obtained from 0.5×FeSSIF and FeSSIF media (P<0.05). But the permeability of API was larger than that of precipitates obtained from biorelevant media. As shown in Figure 8 B, only the precipitate obtained from 0.5×FeSSIF medium had different permeability from other samples when the donor chamber contained pH 5.0 PBS (P<0.05).”

Comment 2: The introduction needs to be concise.

Reply: As required, we have rewritten the “Introduction” section, please see the latest manuscript.

Comment 3: Abstract and conclusion should include a sentence proposing the future direction of the present research.

Reply: Thanks for your advice. According to your suggestion, we have added relevant description of the future research direction in the “Abstract” and the “Conclusion” sections.

Comment 4: Also, the commercial aspects of how precipitations obtained from bio-relevant media industries can benefit.

Reply: The physicochemical properties of precipitates have some differences from API, which may influence the absorption. Therefore, the absorption of re-dissolved precipitate should pay more attention in the process of new drug development. In addition, precipitates obtained from biorelevant media have a higher dissolved rate than that of API. This is one point to improve the dissolved rate of insoluble compounds. All of these can bring some benefits to commercial aspects.

Comment 5: Most of the methodology lacks citing the references from where the technique was adopted and used to conduct the study. The below are a few relevant article suggestions that authors are recommended to cite in the methodology, results /discussion section to justify and strengthen the research data obtained.

International journal of pharmaceutics 511, no. 1 (2016): 680-687.

Gels 7, no. 3 (2021): 96

International journal of pharmaceutics 453, no. 1 (2013): 25-35.

International journal of nanomedicine 10 (2015): 321.

Journal of Pharmacy and Pharmacology 63, no. 3 (2011): 333-341.

Current drug delivery 13, no. 2 (2016): 211-220.

Reply: Thanks for your comments. In the new version, we have cited relevant references in the Methodology, Results, Discussion sections to enhance the persuasion and reliability of the experimental data.

It is our hope that our responses to reviewers’comments and modifications on the manuscript meet the standard for publication of our paper in your esteemed journal. Thank you for the opportunity.

Sincerely,

Xijing Chen

Professor

School of Basic Medicine and Clinical Pharmacy, China Pharmaceutical University, Nanjing, 211198, China.

Email: chenxj-lab@hotmail.com

Reviewer 2 Report

The article entitled Characterizing the physicochemical properties of precipitations obtained from biorelevant media. Is an interesting work I have some suggestion which could be useful to improve the manuscript before it could be accept

  1. The Author mentions that weakly basic compounds show poor solubility but could be interesting mention some of this drugs with references.
  2. Mention in the introduction what is Biorelevant powders, if this is the same that biorelevant media, use the same term during the manuscript is important mention what they content in the introduction, because at the end of the manuscript in the conclusion the author mention that bile salts influence in the solubility importantly.
  3. In addition, is important to mention which is the difference between the different biorelevant media used in this study, if this modify the physicochemical parameters, which are the chemical characteristic os each media employed
  4. Furthermore, is important to mention the concentrations at which the drug employed has effect and if is known at which concentration start to precipitate. This information could be correlated with the concentrations employed in this work. In addition could be more  specific with the concentration employed
  5. The subtitles of results should indicated the results and not the technique employed
  6. In the section results of X-ray diffraction was taken the text of manuscript as the text of the figure 1, this section is confusing.
  7. Change the labels of the Figure 2, because these are in red and is difficult to read.
  8. In the results of microscopy describe which is the importance of different size and morphology of the crystals.

Author Response

Dear Reviewer,

Thank you for your letter and comments concerning our manuscript entitled “Characterizing the physicochemical properties of two weakly basic insoluble drugs and the precipitates obtained from biorelevant media” (Pharmaceutics-1520066). Those comments are all valuable and very helpful for improving our paper. We have carefully studied the comments and have made appropriate corrections in line with the recommendations of reviewers. Changes are highlighted with yellow background in the revised version. The main corrections in the manuscript and the responses to the reviewers’ comments are as follows:

Reviewer 2

The article entitled Characterizing the physicochemical properties of precipitations obtained from biorelevant media. Is an interesting work I have some suggestions which could be useful to improve the manuscript before it could be accept.

Comment 1: The Author mentions that weakly basic compounds show poor solubility but could be interesting mention some of this drugs with references.

Reply: We apologize for the inappropriate expression. We have revised it as “weakly basic insoluble compounds” according to corresponding expression of the published paper. (10.1002/jps.23985; 10.1007/s11095-016-1890-8)

Comment 2: Mention in the introduction what is Biorelevant powders, if this is the same that biorelevant media, use the same term during the manuscript is important mention what they content in the introduction, because at the end of the manuscript in the conclusion the author mention that bile salts influence in the solubility importantly.

Reply: Thanks for your advice. They are different. Biorelevant powder is a commercial product of premixed bile salts and phosphatides that was purchased from Biorelevant.com Ltd., (Surrey, UK). Biorelevant media were prepared by biorelevant powder and buffer solution. We have revised the inappropriate expression in the introduction.

Comment 3: In addition, is important to mention which is the difference between the different biorelevant media used in this study, if this modify the physicochemical parameters, which are the chemical characteristic of each media employed.

Reply: Thanks for your comments. The content of biorelevant powder and the pH of buffer are different in different media. We have added the composition of medium in Table 1 in the new manuscript. The biorelevant media have some influences on the physicochemical parameters of precipitates, such as the changes in crystal structure, surface morphology, melting point, intrinsic dissolution rate and permeability.

Comment 4: Furthermore, is important to mention the concentrations at which the drug employed has effect and if is known at which concentration start to precipitate. This information could be correlated with the concentrations employed in this work. In addition could be more specific with the concentration employed

Reply: Thanks for your valuable comment. We apologize for ignoring the key point. After the solution concentration larger than the supersaturation concentration, nucleation will occur, followed by crystallization and precipitate. As it is difficult to obtain the supersaturation concentration, we referred the equilibrium solubility to estimate the possible concentration that precipitate start to form. We have added solubility of API in various aqueous buffers (pH 1.0 HCl solution, pH 5.0, pH 6.5, and pH 8.0 phosphate buffers) and biorelevant media (FaSSIF and FeSSIF media) in Table 2 of the new manuscript. The details are as follows:

pH 1.0 HCl

(μg/mL)

pH 5.0 PBS

(μg/mL)

pH 6.5 PBS

(μg/mL)

pH 8.0 PBS

(μg/mL)

FaSSIF (μg/mL)

FeSSIF (μg/mL)

Rivaroxaban

6.84 (±0.2)

4.85 (±0.11)

5.85 (±0.02)

5.87 (±0.04)

6.17 (±0.13)

10.8 (±0.17)

Drug-A

42.9 (±0.50)

11.9 (±0.12)

11.6 (±0.16)

11.2 (±0.15)

12.3 (±0.30)

13.8 (±0.85)

Comment 5: The subtitles of results should indicated the results and not the technique employed

Reply: Thanks for your advice. We have revised them according to your suggestion.

Comment 6: In the section results of X-ray diffraction was taken the text of manuscript as the text of the figure 1, this section is confusing.

Reply: As required, we have revised this part in the latest version. Thanks you for your comment again.

Comment 7: Change the labels of the Figure 2, because these are in red and is difficult to read.

Reply: Thanks for your advice. We have revised it. Please see the following picture.

Comment 8: In the results of microscopy describe which is the importance of different size and morphology of the crystals.

Reply: Thanks for your advice. According to your advice, we have rewritten this section and described the importance of different size and morphology for the different crystals.

It is our hope that our responses to reviewers’ comments and modifications on the manuscript meet the standard for publication of our paper in your esteemed journal. Thank you for the opportunity.

Sincerely,

Xijing Chen

Professor

School of Basic Medicine and Clinical Pharmacy, China Pharmaceutical University, Nanjing, 211198, China.

Email: chenxj-lab@hotmail.com

Reviewer 3 Report

This manuscript reports the results on characterizing the physicochemical properties of precipitations obtained from biorelevant media. The objective of this study is reasonable and the reviewer also agree with the necessity of this study. However, as indicated by the abstract, any data to optimize physiologically based pharmacokinetic (PBPK) model with the model drug was not provided and supported by their observation. In addition, overall manuscript seems to be poorly described and discussed with their observation. Also, there are still some critical issues and suggestions to improve the manuscript as below.

  1. One of the weak point is about the model drug selections. Authors try to choose weak base drugs as model drugs. However, there is no physiochemical information such as pKa, log P, solubility about rivaroxaban and Drug-A. Do these drug are weak base drugs? If the authors want to demonstrate their interested data for weak base drugs, the model drugs should be rationalized and the number of model drugs should be enough. Since there is no information about Drug-A, the rationale to choose as weak base drugs remains unclear.
  2. One of the weak point is the absence of any PBPK optimization data to improve accurate oral absorption with weak base drugs as model drugs. Because the manuscript includes so much description about PBPK modeling in the introduction and discussion section. To demonstrate authors’s conclusion about PBPK model application, the further study should be conducted in this study.
  3. The reason of differences between rivaroxaban and Drug-A on the various physicochemical properties is recommended to describe in detail.
  4. In the method section, which is Preparation, the composition and pH of biorelevant medias should be described in detail. These biorelevant medias contain intestinal digestion enzymes such as bile acid and pancreatic juice.
  5. Title should include the weak base drugs.

Author Response

Dear Reviewer,

Thank you for your letter and comments concerning our manuscript entitled “Characterizing the physicochemical properties of two weakly basic insoluble drugs and the precipitates obtained from biorelevant media” (Pharmaceutics-1520066). Those comments are all valuable and very helpful for improving our paper. We have carefully studied the comments and have made appropriate corrections in line with the recommendations of reviewers. Changes are highlighted with yellow background in the revised version. The main corrections in the manuscript and the responses to the reviewers’ comments are as follows:

Reviewer 3

This manuscript reports the results on characterizing the physicochemical properties of precipitations obtained from biorelevant media. The objective of this study is reasonable and the reviewer also agree with the necessity of this study. However, as indicated by the abstract, any data to optimize physiologically based pharmacokinetic (PBPK) model with the model drug was not provided and supported by their observation. In addition, overall manuscript seems to be poorly described and discussed with their observation. Also, there are still some critical issues and suggestions to improve the manuscript as below.

Comment 1: One of the weak point is about the model drug selections. Authors try to choose weak base drugs as model drugs. However, there is no physiochemical information such as pKa, log P, solubility about rivaroxaban and Drug-A. Do these drug are weak base drugs? If the authors want to demonstrate their interested data for weak base drugs, the model drugs should be rationalized and the number of model drugs should be enough. Since there is no information about Drug-A, the rationale to choose as weak base drugs remains unclear.

Reply: Thanks for your valuable comments. They are weakly basic insoluble drugs. According to your advice, we have added the physiochemical information such as pKa, log P of rivaroxaban and Drug-A in the “Material and method” section. The details as follows: “Rivaroxaban (pKa 13.6, log P 1.9, purity≥98%) and Drug-A (pKa 10.41, log P 0.93, purity≥98%) were selected as the weakly basic model drugs in this manuscript”. In addition, we have added the solubility of API in various aqueous buffers (pH 1.0 HCl solution, pH 5.0, pH 6.5, and pH 8.0 phosphate buffers) and biorelevant media (FaSSIF and FeSSIF media) in Table 2 of the new manuscript. According to the solubility and the pKa, it can fully show that Rivaroxaban and Drug-A are weakly basic drugs.

Comment 2: One of the weak points is the absence of any PBPK optimization data to improve accurate oral absorption with weak base drugs as model drugs. Because the manuscript includes so much description about PBPK modeling in the introduction and discussion section. To demonstrate authors’s conclusion about PBPK model application, the further study should be conducted in this study.

Reply: Thanks for your valuable comments. Systematic study of the physicochemical properties of precipitates obtained from biorelevant media in vitro is the prerequisite for understanding the absorption mechanism of re-dissolved precipitates. In this study, we are focusing on comparing the difference between precipitate. In order to highlight research results, we have deleted too much description of PBPK in the introduction and discussion sections. Modeling is another work, and this work is full of great challenges because it needs a lot of data to support. Relevant research of PBPK will become the focus of our next work.

Comment 3: The reason of differences between rivaroxaban and Drug-A on the various physicochemical properties is recommended to describe in detail.

Reply: According to your advice, we have added relevant reason for the differences in physicochemical properties between rivaroxaban and Drug-A in the new version.

Comment 4: In the method section, which is Preparation, the composition and pH of biorelevant media should be described in detail. These biorelevant media contain intestinal digestion enzymes such as bile acid and pancreatic juice.

Reply: Thanks for your advice. The composition of biorelevant media was shown in Table 1 of the new manuscript. We have added relevant information in the “Materials and Methods” section.

Comment 5: Title should include the weak base drugs.

Reply: Accepted. We have revised it in the latest version.

It is our hope that our responses to reviewers’comments and modifications on the manuscript meet the standard for publication of our paper in your esteemed journal. Thank you for the opportunity.

Sincerely,

Xijing Chen

Professor

School of Basic Medicine and Clinical Pharmacy, China Pharmaceutical University, Nanjing, 211198, China.

Email: chenxj-lab@hotmail.com

Reviewer 4 Report

General Comments to the Authors on the Manuscript: The authors explore in vitro experimentation to predict in-vivo oral PK. The authors use two model drugs rivaroxaban and Drug-A, and experimentally isolate precipitates from aqueous-based biorelevant media and then characterize the precipitates with the goal of predicting oral human PK. The authors discuss in terms of weakly basic drugs, but rivaroxaban is non-ionizable, so the authors should explain why they choose rivaroxaban as a model drug.

Specific Comments to the Authors on the Manuscript:

  1. Grammar: Throughout the manuscript and in the title the authors use the term “precipitation” when referring to precipitates that the authors isolated, thus suggest to change “precipitation” when used as a noun to “precipitates”.
  2. Why did the authors choose rivaroxaban as a model drug? Drug-A is a weak base, but rivaroxaban has no ionizable group.
  3. The authors should report the pKa of their Drug-A.
  4. Report the HPLC method.
  5. Lines 63-72: This is a good paragraph that nicely explains the theory
  6. Line 94-95: Materials: Biorelevant powders, were these the FeSIF and FaSIF?
  7. The authors should report the difference between the FaSIF and FeSIF.
  8. Lines 158-165 Mini-Intrinsic dissolution: Was the volume really 10 mL only? Thus 1 mg/mL (used 10 mg).
  9. Table 1: limit significant figures to 2 or 3 not 4
  10. Figure 1: Why did the 0.5Fe of rivaroxaban have a new peak near 10 deg.? Why did all the precipitates from Drug-A different pattern than Drug-A API?
  11. Figure 4: Why are the DCS traces of Drug-A precipitates different when the XRPDs in Figure 1 all look the same?
  12. Lines 261-262: “… newly formed carbon-oxygen double bond after the interaction between Drug-A and acetic acid…”, but this seems quite unlikely.
  13. Lines 270-287 and Figure 5: Do not confuse solubility with rate of dissolution.
  14. Figure 5: the most important data, limit x-axis 0 to 12 hours.
  15. The authors do not present a plot of intrinsic dissolution….concentration as a function of time. It would be useful to have an example plot of intrinsic dissolution.
  16. Line 183-184: Permeability: why use excess solid as opposed to a known solution concentration? Refer the reader to lines 392-394 for your reason for using solid.
  17. Why was the permeability reduced when using SIF? Is the effect of micelles?
  18. Figures 6 and 7: confusing due to no x-axis, and too much data. Suggest bar charts
  19. Awkward sentences:
    1. Line 18: “…absorption of precipitation…” suggest change to “…absorption of redissolved precipitate…”
    2. In two places the authors state that the chemical structure can be affected by intermolecular forces or biorelevant media, but this is scientifically not correct.
      1. Line 26: “…the molecule structure of model drug (Drug-A) can also be affected by the intermolecular forces.”, I think the authors are trying to say “…the crystal lattice of the precipitated model drug (Drug-A) can also be affected by the intermolecular forces.”
      2. Similarly Line 436: “…biorelevant media can affect the molecular structure of a certain compound…”, I think the authors are trying to say “…biorelevant media can affect the crystal lattice of a precipitated compound...”
    3. Line 31: “Therefore, compounds will have…” suggest to change to “Therefore, it is predicted that compounds will have…”
    4. Line 63: “Generally, weakly basic drugs have the complex process in the GI tract”, suggest to change to something like “Generally, weakly basic drugs in the GI tract can undergo complex processes involving dissolution, supersaturation, precipitation, redissolution, and absorption.”
    5. Line 90: “…absorption of precipitation…” suggest change to “…absorption of redissolved precipitate…
    6. Line 345: ”…under…, is this “…understand…”?.

Author Response

Dear Reviewer,

Thank you for your letter and comments concerning our manuscript entitled “Characterizing the physicochemical properties of two weakly basic insoluble drugs and the precipitates obtained from biorelevant media” (pharmaceutics-1520066). Those comments are all valuable and very helpful for improving our paper. We have carefully studied the comments and have made appropriate corrections in line with the recommendations of reviewers. Changes are highlighted with yellow background in the revised version. The main corrections in the manuscript and the responses to the reviewers’ comments are as follows:

Reviewer 4

Comment 1: General Comments to the Authors on the Manuscript: The authors explore in vitro experimentation to predict in-vivo oral PK. The authors use two model drugs rivaroxaban and Drug-A, and experimentally isolate precipitates from aqueous-based biorelevant media and then characterize the precipitates with the goal of predicting oral human PK. The authors discuss in terms of weakly basic drugs, but rivaroxaban is non-ionizable, so the authors should explain why they choose rivaroxaban as a model drug.

Reply: Thanks for your comment. We have explained the reason for selecting rivaroxaban as model drug in new manuscript. Although rivaroxaban is a non-ionizing compound, it has one secondary amine groups in its structural formula, and the pKa of rivaroxaban is about 13.6. It is a weakly basic drug.

Comment 2: Grammar: Throughout the manuscript and in the title the authors use the term “precipitation” when referring to precipitates that the authors isolated, thus suggest to change “precipitation” when used as a noun to “precipitates”.

Reply: Accepted. We have revised it in the new manuscript.

Comment 3: Why did the authors choose rivaroxaban as a model drug? Drug-A is a weak base, but rivaroxaban has no ionizable group.

Reply: Thanks.

The reason for choosing rivaroxaban as a model drug is mainly based on:(1) There is one secondary amine groups in its structural formula; (2) The solubility of rivaroxaban is 6.84 μg/mL in pH 1.0 HCl solution and 5.24 μg/mL in pH 8.0 phosphate buffer; (3) The pKa of rivaroxaban is about 13.6.

Drug A also has one secondary amine groups in the structural formula, the pKa of Drug A is about 10.41, and the solubility of Drug-A in pH 1.0 HCl is 42.93 ug/mL, and 11.25 ug/mL in pH 8.0 phosphate buffer. Therefore, rivaroxaban and Drug-A are weakly basic insoluble compounds.

In the new manuscript, we have added pKa, log P and purity of the model drugs in “Material” section, and added the solubility of API in various aqueous buffers (pH 1.0 HCl solution, pH 5.0, pH 6.5, and pH 8.0 phosphate buffers) and biorelevant media (FaSSIF and FeSSIF media) in Table 2 of the new manuscript.

Comment 4: The authors should report the pKa of their Drug-A.

Reply: As required, we have added the pKa in the new manuscript. The details as follows: “Rivaroxaban (pKa 13.6, log P 1.9, purity≥98%) and Drug-A (pKa 10.41, log P 0.93, purity≥98%) were selected as the weakly basic model drugs in this work”.

Comment 5: Lines 63-72: This is a good paragraph that nicely explains the theory

Reply: Thank you so much.

Comment 6: Line 94-95: Materials: Biorelevant powders, were these the FeSIF and FaSIF?

Reply: The Biorelevant powders are not FeSIF and FaSIF. Biorelevant powder is a commercial product of premixed bile salts and phosphides. Biorelevant media were prepared by a certain amount biorelevant powders and buffer solution.

Comment 7: The authors should report the difference between the FaSIF and FeSIF.

Reply: Thanks for your advice. In the new version, we have added the difference between FaSIF and FeSIF including the composition as displayed in Table 1. The mainly difference is the content of biorelevant powder, the pH of the finial medium and the composition of buffer solution. Please see Table 1 in the new manuscript.

Comment 8: Lines 158-165 Mini-Intrinsic dissolution: Was the volume really 10 mL only? Thus 1 mg/mL (used 10 mg).

Reply: Yes, the volume is 10 mL. Since the equilibrium solubility of rivaroxaban and Drug-A were no more than 15 μg/mL in pH 5.0 PBS, pH 6.5 PBS, FaSSIF medium and FeSSIF medium, the dissolved compound (real concentration) in the experiment of IDR is very small. We have displayed concentration-time curve in Figure 6 in the new manuscript.

Comment 9: Table 1: limit significant figures to 2 or 3 not 4.

Reply: Accepted. We have revised the limit significant figures to 3 in Table 1 (Table 3 in the new manuscript).

Comment 10: Figure 1: (1) Why did the 0.5Fe of rivaroxaban have a new peak near 10 deg.? (2) Why did all the precipitates from Drug-A different pattern than Drug-A API?

Reply: Thanks! (1) Actually, 0.5 Fe, FaB, FeB, Fe and 2 Fe all have the peak near 10 degree, which was mainly caused by the phosphate buffer solution.

(2) For the second question, the reason can be explained as follows: the pattern of API is similar to API-ACN, FaB, FeB and Fa, as the peak sites is almost the same. However, the pattern of API is different from that of 0.5 Fe, Fe and 2 Fe, which was attributed to the addition of large amount of biorelevant powder.

Comment 11: Figure 4: Why are the DCS traces of Drug-A precipitates different when the XRPDs in Figure 1 all look the same?

Reply: Thanks! Differential scanning calorimetry (DSC) is a thermal analysis technique that investigates changes in heat capacity of compounds as a function of temperature. DSC reports phase transition in the samples, which are enthalpies absorbed (endothermic) or released (exothermic) during the transition(10.1080/10408398.2012.734343). Figure 4 presents the exothermic peaks.

XRD report the difference of crystal structure, and some different crystal phases may correspond to the same crystal peak. So DSC is different and XRD looks the same.

Comment 12: Lines 261-262: “… newly formed carbon-oxygen double bond after the interaction between Drug-A and acetic acid…”, but this seems quite unlikely.

Reply: You are right, the original description is not appropriate, so we revised relevant description in the new manuscript.

Comment 13: Lines 270-287 and Figure 5: Do not confuse solubility with rate of dissolution.

Reply: As required, we have modified related content to avoid the aforementioned confusion.

Comment 14: Figure 5: the most important data, limit x-axis 0 to 12 hours.

Reply: According to your suggestion, we have revised the X-axis in the new manuscript. 

Comment 15: The authors do not present a plot of intrinsic dissolution… concentration as a function of time. It would be useful to have an example plot of intrinsic dissolution.

Reply: Thanks for your advice! We have added an example plot of intrinsic dissolution in the Figure 6.

Comment 16: Line 183-184: Permeability: why use excess solid as opposed to a known solution concentration? Refer the reader to lines 392-394 for your reason for using solid.

Reply: Thanks. In order to simulate the re-dissolution progress of precipitate in the GI tract, a slight excess of crystalline powder was added into the donor chamber. We have added relevant reason in the part of method in the new manuscript.

Comment 17: (1) Why was the permeability reduced when using SIF? (2) Is the effect of micelles?

Reply: Thanks. (1) After the binding between the compound and the bile salts, the increasing molecular size and the hydrated layer will prevent the compounds from passing through the lipophilic permeable membrane. Hence, the permeability was when using SIF. (2) It is the binding between compounds and bile salts. Although biorelevant media have good solubilization ability, the amount of bile salts has the limited capacity to dissolve the excess compound in the donor chamber. Therefore, there is little capacity of bile salts to form the micelles. In the new manuscript, we have added the corresponding description: “Although biorelevant media have good solubilization ability, the amount of bile salts has the limited capacity to dissolve all compounds in the donor chamber. Theoretically, the hydrophobic groups of compounds can be wrapped by bile salts, and the hydrophilic groups are exposed to the medium. After the binding between the compound and the bile salts, the increasing molecular size and the hydrated layer will prevent the compounds from passing through the lipophilic permeable membrane.”

Comment 18: Figures 6 and 7: confusing due to no x-axis, and too much data. Suggest bar charts

Reply: Thanks for your advice! We used the bar charts in the new manuscript.

Comment 19: Awkward sentences:

    1. Line 18: “…absorption of precipitation…” suggest change to “…absorption of redissolved precipitate…”
    2. In two places the authors state that the chemical structure can be affected by intermolecular forces or biorelevant media, but this is scientifically not correct.
      1. Line 26: “…the molecule structure of model drug (Drug-A) can also be affected by the intermolecular forces.”, I think the authors are trying to say “…the crystal lattice of the precipitated model drug (Drug-A) can also be affected by the intermolecular forces.”
      2. Similarly Line 436: “…biorelevant media can affect the molecular structure of a certain compound…”, I think the authors are trying to say “…biorelevant media can affect the crystal lattice of a precipitated compound...”
    3. Line 31: “Therefore, compounds will have…” suggest to change to “Therefore, it is predicted that compounds will have…”

Reply: Thanks so much!

    1. Line 63: “Generally, weakly basic drugs have the complex process in the GI tract”, suggest to change to something like “Generally, weakly basic drugs in the GI tract can undergo complex processes involving dissolution, supersaturation, precipitation, redissolution, and absorption.”
    2. Line 90: “…absorption of precipitation…” suggest change to “…absorption of redissolved precipitate…
    3. Line 345: ”…under…, is this “…understand…”?.

Reply: Thanks for your thorough reading, we really appreciate you. According to your suggestion, we have revised these sentences in the new version. Meanwhile, we have also checked the whole manuscript to polish the language.

Thanks to you again.

Sincerely,

Xijing Chen

Professor

School of Basic Medicine and Clinical Pharmacy, China Pharmaceutical University, Nanjing, 211198, China.

Email: chenxj-lab@hotmail.com

Reviewer 5 Report

the present report is very interesting and it deserves the publication in the journal. However, some minor commetns to address are reportes below in order to improve the quality of the paper.

-Rivaroxaban and Drug-A purity should be reported

-in Table 1 the number of the samples considered should be reported

-row 215 the style is not correct please format correctly

-implication of this study should be clearly highlighted as well as the limitation of the study

Author Response

Dear Reviewer,

Thank you for your letter and comments concerning our manuscript entitled “Characterizing the physicochemical properties of two weakly basic insoluble drugs and the precipitates obtained from biorelevant media” (pharmaceutics-1520066). Those comments are all valuable and very helpful for improving our paper. We have carefully studied the comments and have made appropriate corrections in line with the recommendations of reviewers. Changes are highlighted with yellow background in the revised version. The main corrections in the manuscript and the responses to the reviewers’ comments are as follows:

Reviewer 5

The present report is very interesting and it deserves the publication in the journal. However, some minor commetns to address are reportes below in order to improve the quality of the paper.

Comment 1: Rivaroxaban and Drug-A purity should be reported

Reply: Thanks for your advice. We have added the purity of Rivaroxaban and Drug-A in the “Materials and Method” section in the new version.

Comment 2: in Table 1 the number of the samples considered should be reported

Reply: As required, we have added the number of the samples in Table 1 (Table 3 in the new manuscript).

Comment 3: row 215 the style is not correct please format correctly

Reply: Thank you so much. We have revised it in the new version.

Comment 4: implication of this study should be clearly highlighted as well as the limitation of the study

Reply: Thanks for your valuable comment. The implication and the limitation of this study have been highlighted in the last paragraph of the “Discussion” section in the new manuscript.

Sincerely,

Xijing Chen

Professor

School of Basic Medicine and Clinical Pharmacy, China Pharmaceutical University, Nanjing, 211198, China.

Email: chenxj-lab@hotmail.com

Round 2

Reviewer 1 Report

chnages made by author- accept

Reviewer 2 Report

I recommend the publication of the article due to its has been  sufficiently improved.

Reviewer 3 Report

The revised manuscript tries to reflect the reviewer's suggestions and concerns to some extent. 

Reviewer 5 Report

My previous concerns have been addressed.